# Pathogenic Variants and Genotypes of the *CFTR* Gene in Russian Men with Cystic Fibrosis and CBAVD Syndrome

**DOI:** 10.3390/ijms242216287

**Published:** 2023-11-14

**Authors:** Vyacheslav Chernykh, Stanislav Krasovsky, Olga Solovova, Tagui Adyan, Anna Stepanova, Ekaterina Marnat, Maria Shtaut, Anna Sedova, Tatyana Sorokina, Tatyana Beskorovainaya, Elena Kondratyeva, Olga Shchagina, Aleksandr Polyakov

**Affiliations:** 1Research Centre for Medical Genetics, 115522 Moscow, Russia; sa_krasovsky@mail.ru (S.K.); olga_pilyaeva@list.ru (O.S.); adyan@dnalab.ru (T.A.); cany@yandex.ru (A.S.); shtaut@yandex.ru (M.S.); luoravetlanka@gmail.com (A.S.); reprolab@med-gen.ru (T.S.); t-kovalevskaya@yandex.ru (T.B.); elenafpk@mail.ru (E.K.); schagina@med-gen.ru (O.S.); polyakov@med-gen.ru (A.P.); 2Pirogov Russian National Research Medical University of the Ministry of Healthcare of the Russian Federation, 117997 Moscow, Russia; marnat_eg@rsmu.ru

**Keywords:** cystic fibrosis, CBAVD syndrome, gene variant, CFTR, azoospermia

## Abstract

Pathogenic *CFTR* variants cause cystic fibrosis (CF), and CF-related disorders (CF-RD), including bilateral aplasia of the vas deferens (CBAVD). The spectrum of clinical manifestations depends on the *CFTR* genotype. The frequency and spectrum of the *CFTR* variants vary between populations and clinical groups. *CFTR* variants and genotypes were analyzed in Russian men with CF (*n* = 546) and CBAVD syndrome (*n* = 125). Pathogenic variants were detected in 93.95% and 39.2% of the CF and CBAVD alleles, respectively. The most frequent c.1521_1523del (F508del; p.Phe508del) variant was found in 541 (49.5%) CF alleles. A total of 162 *CFTR* genotypes were revealed in CF patients, including 152 homozygous and 394 compound-heterozygous. The most common CF-genotype was F508del/F508del (24.9%). Other frequent CF-genotypes were F508del/3849+10kbC>T, F508del/CFTRdele2,3, and F508del/E92K. CF-causing variants and/or 5T allele were found in 88% of CBAVD patients: 5T/CFTRmut (48.0%), CFTRmut/N (17.6%), CFTRmut/CFTRmut (6.4%), 5T/5T (10.4%), 5T/N (5.6%) and N/N (12.0%), with the most common CBAVD-genotype being F508del/5T (29.6%). The allele frequencies of F508del, CFTRdele2,3 394delTT, and 3849+10kbC>T were significantly higher in CF patients. L138ins/L138ins, 2184insA/E92K, and L138ins/N genotypes were found in CBAVD, but not in CF patients. The results indicate certain differences in the frequency of some *CFTR* variants and genotypes in Russian CF and CBAVD patients.

## 1. Introduction

Infertility is a common medico-social problem affecting 10–15% of married couples. “Male factor” is revealed in about half of all infertile couples, with genetic abnormalities responsible for at least 50% of cases of severe infertility [1]. Common genetic causes of male infertility include sex chromosome abnormalities (Klinefelter syndrome, X/XY mosaicism, disomy Y, and others), the Y chromosome microdeletions in the AZF (azoospermia factor) locus, and *CFTR* gene variants [2,3,4].

Pathogenic *CFTR* (cystic fibrosis transmembrane conduction regulator) gene variants are a cause of cystic fibrosis, CF (OMIM#219700), and CFTR-related disorders (CFTR-RD), particularly, congenital bilateral aplasia of vas deferens (CBAVD) [2,3,4,5]. Cystic fibrosis is a common monogenic autosomal recessive disorder in Caucasian populations. This multisystemic disease results from impaired sodium and chloride transport across epithelial surfaces, and is characterized by progressive obstructive pulmonary disorder, pancreatic, hepatobiliary, and gastrointestinal dysfunction [3,4].

CBAVD syndrome (OMIM# 277180) is an autosomal recessive disease, considered as “genital form” of CF in male patients. In adult male patients, CBAVD is one of the manifestations of Cystic Fibrosis, but it may be independent of CF disease without extragenital symptoms [4]. Some patients with CBAVD syndrome show extragenital symptoms of CF (chronic pancreatitis, bronchitis, sinusitis), which are usually mild and develop mainly in adulthood. Therefore, differential diagnosis and identification of mild (“atypical”) forms of CF and other CFTR-RD in azoospermic patients, especially the heterozygosity for *CFTR* gene variants, is highly relevant. CBAVD is a common cause of obstructive azoospermia, accounting for 25% of its cases, and is diagnosed in 1–2% of infertile men from various populations [6,7]. However, the actual frequency of congenital bilateral and unilateral aplasia of vas deferens (CBAVD and CUAVD, respectively) is rather underestimated. It is due to a number of problems in differential diagnosis CAVD from other forms of obstructive azoospermia and oligozoospermia [8]. Overall, in 80% patients, CBAVD syndrome is resulted from CF-causing variants/5T allele of the *CFTR* gene; rare cases of this disorder are caused by pathogenic variants in the *ADGRG2* or other candidate genes (*SLC9A3*, *SCNN1B*, and *CA12*) [8].

The *CFTR* gene is located on chromosome 7 (7q31.2), has a size of about 189 Kb, and consists of 27 exons and 26 introns [9]. CFTR protein is a member of the ATP-binding cassette (ABC) transporter superfamily, which functions as a chloride channel, controlling ion and water secretion and absorption in epithelial tissues [9]. To date, more than 2,000 pathogenic variants in the *CFTR* gene have been identified [9]. The severity and spectrum of clinical manifestations depend on the severity of the *CFTR* genotype, which determines the degree of defect/preservation of the encoded protein.

Numerous studies show that the genotypes of patients with CF and CBAVD syndrome differ significantly, with the combination of two “severe” pathogenic variants in trans-position, i.e., on both *CFTR* alleles, leads to severe forms of CF [10]. Commonly, CBAVD syndrome is characterized by genotypes with “mild” *CFTR* mutations in combination with the 5T allele (IVS-9T5 polymorphic variant), which is detected in 40–50% of CBAVD patients and in 4–5% individuals of Caucasian origin [11,12]. The frequency of CBAVD syndrome varies in different regions (in European populations, it is slightly higher than in Asian populations), which is probably due to the different frequencies of pathogenic *CFTR* gene variants in different regions and ethnic groups [10,11].

The frequency and spectrum of the *CFTR* gene variants significantly vary between different populations, ethnic and clinical groups, particularly CF and CF-RD. The spectrum of the *CFTR* gene variants and genotypes in Russian CF patients has been studied in detail, including differences in various age groups (children and adults), as well as in different regions of the Russian Federation [13]. However, to date, no combined study of Russian CF patients and men with CBAVD syndrome has been performed. Therefore, the aim of the present study was a comparative analysis of common pathogenic *CFTR* gene variants and genotypes in Russian men with cystic fibrosis and Russian patients with CBAVD syndrome.

## 2. Results

CF-causing variants were detected in 1124 (83.76%) of 1342 analyzed *CFTR* alleles, including all of 546 (100%) CF patients and 90 (72%) men with CBAVD syndrome. In CF patients, pathogenic CF-causing CFTR gene variants were found in 1026 (93.95%) of the 1092 analyzed alleles. As many as 91 different *CFTR* gene variants were revealed in CF patients. Sixteen different CF-causing variants (F508del, CFTRdele2,3, 3849+10kbC>T, E92K, 2184insA, 2143delT, L138ins, N1303K, G542X, W1282X, 1677delTA, R334W, 604insA, G85E, 4015delA, 3272-26A>G) were detected in both CF patients and CBAVD patients (Table 1).

In the CF patients’ group, 162 *CFTR* genotypes were detected. CF patients were found to be homozygous (*n* = 152) or compound-heterozygous (*n* = 394). The most frequent was the F508del/F508del genotype revealed in 136 (24.9%) CF patients (Table 2).

The second most common genotype was compound heterozygous F508del/unknown, detected in 45 (8.2%) CF patients and 45 (36.0%) CBAVD patients. Other three common CF-causing genotypes (F508del/3849+10kbC>T, F508del/CFTRdele2,3 and F508del/E92K) F508del/3849+10kbC>T, F508del/CFTRdele2,3 and F508del/E92K were found in 38 (7.0%), 33 (6.0%), and 23 (4.2%) CF patients, respectively.

In CBAVD patients, pathogenic CF-causing variants were detected in 98 (39.2%) of *CFTR* alleles. 16 different variants were revealed in this group (Table 1). Eight individuals with CBAVD syndrome were found to have two CF-causing variants, including two compound-heterozygous (*n* = 7) and one homozygous (L138ins/L138ins) (Table 3).

In CBAVD patients, the F508del variant was revealed in 45 (18%) of 250 alleles, in 36% of individuals. In this group, besides F508del mutations, more frequent *CFTR* gene variants were L138ins, CFTRdele2,3(21Kb), W1282X, which consisted of 8.2%, 7.1%, and 6.1% of all detected CF-causing variants, respectively. Overall, pathogenic *CFTR* gene variants and/or 5T allele were found in 110 (88.0%) of the 125 examined CBAVD patients (Figure 1).

Analysis of the IVS9-Tn polymorphic locus in intron 9 of the *CFTR* gene detected three allelic variants: 5T, 7T, and 9T. The 5T (IVS9-5T) allele was found in 93 (37.2%) of 250 alleles, in 80 of 125 (64%) patients with CBAVD syndrome (Table 3). This allelic variant of the *CFTR* gene was detected in the homozygous state (*n* = 13; 10.4% of patients) and in the heterozygous state (*n* = 67; 53.6% of patients) in combination with 7T or 9T alleles. Totally, *CFTR* gene mutation and/or 5T were found in 110 (88.0%) of 125 CBAVD patients (Table 3). Following *CFTR* genotypes were revealed in CBAVD patients: 5T/mut (*n* = 60; 48.0%), mut/N (*n* = 20; 17.6%), mut/mut (n = 8; 6.4%), 5T/5T (n = 13; 10.4%), 5T/N (n = 7; 5.6%), and N/N (n = 15; 12.0%), at that the most common was the F508del/5T genotype detected in 45 (36.0%) individuals in this group (Figure 1).

The allelic frequency of some *CFTR* variants dramatically differs between the patient groups due to the high prevalence of homozygosity and compound-heterozygosity in CF patients. So, the c.1521_1523delCTT (p.(Phe508del), F508del)) variant was a more common variant in both CF (44.4%) and CBAVD (20.4%) patients, but the allelic frequency (AF) was significantly higher in the former group (0.49542 and 0.2, respectively; *χ*_2_ = 70.853; *p* < 0.001). Also, three other variants (CFTRdele2,3, 3849+10kbC-T, and 394delTT) have shown statistically significant higher (*p* < 0.003) allele frequencies in CF patients than in men with CBAVD syndrome (Table 1). The 5T allele was found accidentally in one patient who developed pancreas-sufficient CF (PS-CF) with R117C(c.349C>T)/IVS9-TG12-T5 genotype.

Some *CFTR* genotypes showed a statistically significant (*p* < 0.05) difference between CF and CBAVD patients. So, the prevalence of the F508del/N genotype was significantly higher in CBAVD individuals than in CF patients (36.0% vs. 8.24%; *χ*_2_ = 65.084; *p* < 0.001). F508del/F508del, F508del/3849+10kbC>T, F508del/CFTRdele2,3 and F508del/E92K genotypes were found only in CF patients (Table 2). L138ins/L138ins, 2184insA/E92K, and L138ins/N genotypes were found in men with CBAVD syndrome, but not in CF patients (Table 2 and Table 3).

## 3. Discussion

The results of the present study indicate a high prevalence of *CFTR* gene mutations in Russian men with CBAVD syndrome. A comparative analysis of the prevalence of various *CFTR* pathogenic variants in the two evaluated groups indicates a similar distribution of their occurrence among both adult men with CF and CBAVD syndrome from the Russian Federation. According to our recent research, ten following CF-causing variants, F508del, CFTRdele2,3, L138ins, W1282X, 1677delTA, 3849+10kbC>T, E92K, 2143delT, G542X, and 2184insA, are more common in Russian infertile men [14]. Their allelic frequency (AF) is 0.0005 or more, and the proportion of each variant is 2% or more of all detected pathogenic *CFTR* gene variants in both Russian infertile men and CBAVD patients.

F508del (c.1521_1523delCTT; p.Phe508del) variant is the most common *CFTR* gene mutation both in Russian men with cystic fibrosis and patients with CBAVD syndrome (44.4% and 52.0% of all detected mutations in these groups, respectively). Previously, we examined *CFTR* genotypes in a sample of 72 Russian patients with CBAVD [15]. The data of the evaluation of *CFTR* variants and genotypes in a larger sample of CBAVD patients are consistent with data on a smaller sample of patients.

All examined CF patients are characterized by the presence of one or two pathogenic *CFTR* gene variants, either in a homozygous or in compound heterozygous state. While in the sample of CBAVD patients, two mutations are rare (<5%), CBAVD is characterized by “mild” *CFTR* genotypes [11]. Severe CF-causing variants in the homozygous state were not detected among the examined CBAVD patients. Out of 125 patients, only 8 patients had two CF-causing variants (Table 3). One individual was found to have a mutation in the homozygous state (L138ins/L138ins), and 7 patients were compound-heterozygous for the combination of “severe” (F508del) variants and one of the “mild” *CFTR* gene variants, or two “mild” *CFTR* gene variants. It should be noted that at least some of them may have or develop “atypical” cystic fibrosis with age.

According to the literature, men from European populations with “isolated CBAVD”, CBAVD without diagnosed cystic fibrosis, homozygotes for “severe” pathogenic CF-causing variants were also not revealed, but compound heterozygotes for a “severe” mutation with the R117H (c. 350G>A) variant of the *CFTR* gene are often found in patients with CBAVD from Western European populations [10,11,16]. The R117H variant is a “mild” mutation, the allelic frequency of which in men with CBAVD averages 3%, of which 4.0–11.3% in Caucasian patients, while the F508del/R117H genotype is found in 4–6% of patients. This variant of the *CFTR* gene is not typical for other populations [10,11], including Russian CF patients [17]. Thus, we did not detect any R117H variant among patients from both samples.

Interestingly, 394delTT is one of the common *CFTR* gene variants detected in 17 (3.1%) Russian CF patients (including in homozygous, *n* = 4, and heterozygous, *n* = 13), but not in Russian men with CBAVD syndrome. This variant is the second (4%) of the most frequent “Nordic” *CFTR* mutations in CF patients from Danish, Swedish, Norwegian, and Finnish populations [18]. In West-European populations, this variant has also been reported in a few patients with CBAVD/CUAVD syndromes [8]. According to the RUSeq database [http://ruseq.ru/ accessed on 20 September 2023], in the European part of Russia, the allelic frequency (AF) of the 394delTT (rs121908769) variant is 0.0002134. This variant is the second most common *CFTR* gene variant (3.5% of all detected mutations) in CF patients from Bashkortostan [19]. In participants (*n* = 642) from the population-based cohort study ESSE-Vologda, the 394delTT (rs121908769) variant was found to have 0.0047 alleles [20]. This variant of the *CFTR* gene was found in a large cohort of 6,033 Russian infertile men [14]. However, a heterozygous 394delTT variant was detected in 2 of the 2,146 Russian infertile men examined by Solovyova et al. (2018) [21]. Both of these patients had a 394delTT/N, 7T/9T genotype; one of them was normozoospermic; the spermatological diagnosis of the second patient was not provided; and the presence of CBAVD was not indicated.

Overall, the *CFTR* gene mutation and/or 5T allele was found in 88% of Russian infertile men with CBAVD syndrome. This is consistent with our results and data from other researchers, who found that 80–85% of CBAVD patients of Caucasian origin have CF-causing variants and/or 5T allele of the *CFTR* gene [3]. In the spectrum of *CFTR* genotypes, CFTRmut/5T was the most common in our cohort (29.6%) and in other samples of CBAVD patients, especially in a large cohort of patients from France (28.44%) [3]. The second most common *CFTR* genotype in our cohort of Russian men with CBAVD syndrome was 5T/5T homozygosity, revealed in 10.4% of patients. Recently, we reported the results of an analysis of the *CFTR* gene on a smaller sample of Russian CBAVD patients (*n* = 72) and they were generally similar to the data on a larger sample [15].

Two CF-causing mutations in the *CFTR* genotype were found in 8 (6.4%) patients, while the L138ins variant was revealed in 7 out of 16 alleles, and the proportion of this mutation was 7.1% of all detected mutations. Thus, L138ins was found to be the second most common *CFTR* gene variant in Russian patients with CBAVD syndrome, and the third in prevalence after F508del and CFTRdele2.3(kb) in Russian infertile men. The allele frequency (AF) of this variant in Russian infertile men (*n* = 6033) was 0.0014 [14]. The L138ins variant was found as compound heterozygous F508del/L138ins in four CBAVD patients (Table 3). This genotype was also found in 3 CF patients. Notable, that these CBAVD patients did not have “classic” forms of CF, but at least some had adult-onset pancreatitis or respiratory disease. The possibility of undiagnosed CF or CFTR-RD should be taken into account in the examination of infertile men, especially in patients with obstructive azoospermia.

It should be noted that our study is not without drawbacks. Not all patients underwent sequencing and analysis of major rearrangements or copy number variants, CNVs (for example, deletions and duplications) in the *CFTR* gene. A comprehensive analysis of the nucleotide variants could identify up to 99% of pathogenic alleles. So, Petrova et al. (2018) identified up to 99% and found 10 (9.6%) intergenic CNVs of 104 Russian CF patients. Taulan et al. have identified pathogenic *CFTR* variants (two variants—82.4% and one variant—9.3%) in 91.7% of CBAVD patients from European populations. Subsequently, they have detected two large heterozygous complex deletions (one encompassing exon 2, the other removing exons 22–24) in the *CFTR* gene in two CBAVD patients, and both individuals had pathogenic variants on the other allele [22]. Ma et al. detected 5 CNVs (4 partial deletions and 1 partial duplication) in the *CFTR* gene in 1.9% (5/263) Chinese CBAVD patients [23]. Although their frequency would be low, we cannot exclude rare nucleotide variants or intergenic in some CF and CBAVD patients from our sample, which had *CFTR* alleles considered “normal” or unknown CF-causing variants. Screening for large genomic rearrangements in the *CFTR* gene is beneficial not only in CF but also in CBAVD patients, especially for those azoospermic men who are carriers of a *CFTR* “mild” mutation/5T allele.

The R117H variant is one of the most common non-CF-causing *CFTR* gene variants in European populations and is frequently detected in patients with CBAVD syndrome. This missense variant (p.(Arg117His)) is generally considered to be a mild *CFTR* gene mutation (class IV). In the samples of men with CF and CBAVD syndrome in the present study, the R117H variant was not investigated. According to the RUSeq database [http://ruseq.ru/accessed on 20 September 2023], allelic frequency (AF) of the R117H (rs78655421, NM_000492.4: c.350G>A) variant is 0.0012 in the general population from European part of Russia; in Caucasian and Eastern parts of Russia, the AF of this variant is 0.0000, possibly due to the small number of individuals studied. Solovyova et al. (2018) analyzed the *CFTR* gene in 2146 Russian men from infertile couples, and detected the R117H variant in 8 patients, including four compound-heterozygous F508del/R117H; 9T/7T (azoospermia, *n* = 4); and four heterozygous R117H/N; 7T/7T (azoospermia, *n* = 3; oligozoospermia, *n* = 1) [24]. CBAVD syndrome was diagnosed in 3 patients with the F508del/R117H genotype; the ethnicity of the individuals was not specified. It should be noted that the sample of patients examined by Solovyova et al. consisted of individuals from Western Siberia (Krasnoyarsk). In a cohort of 2146 Russian infertile men examined by Solovieva et al., CBAVD was diagnosed in 10 patients with the following *CFTR* genotypes: F508del/R117H, 9T/7T (*n* = 3), F508del/L138ins, 9T/7T (*n* = 1); F508del/N, 9T/5T-12TG (*n* = 1); CFTRdele2,3(21kb)/N, 7T/5T-12TG (*n* = 1), F508del/N, 9T/7T (*n* = 2), and N/N, 7T/7T (*n* = 2) [24]. No other studies have been performed in Russian patients with CBAVD syndrome.

Modern methods of assisted reproductive technologies (ART), especially the IVF/ICSI procedure, allow having children in patients with various forms of male infertility, including obstructive azoospermia due to CF/CBAVD [25]. The high frequency of the *CFTR* gene variants in patients with CBAVD/obstructive azoospermia should be taken into account; therefore, their children are at increased risk for both CF and CF-RD patients. In this regard, genetic counseling and screening for pathogenic *CFTR* gene variants are very important for these patients and their spouses. Couples at high risk should be offered preimplantation genetic testing (PGT) or prenatal DNA diagnosis.

## 4. Materials and Methods

A cohort of examined individuals consisted of 671 unrelated Russian men, including 125 patients with CBAVD syndrome and 546 adult male CF patients. The patients with CBAVD syndrome had no diagnosed cystic fibrosis. All CF patients had one or two pathogenic *CFTR* gene variants. Informed consent for the study was obtained from all participants.

Molecular study was performed on genomic DNA extracted from peripheral blood leukocytes using the QIAamp DNA Blood Mini Kit (Qiagen, Valencia, CA, USA).

The *CFTR* gene was analyzed for 22 following common in Russian populations pathogenic variants (CFTRdele2,3(21kb), 394delTT, 3944delTG, L138ins, R334W, F508del, I507del, 1677delTA, G542X, 2143delT, 2184insA, 3821delT, 3849+10kb C>T, 604insA, 621+1G>T, E92K, S1196X, W1282X, N1303K, 4022insT, 4015delA, and 3272-26A>G). These variants were detected using amplified fragment length polymorphism (APLF-PCR) and multiplex ligation-dependent probe amplification (MLPA) methods. In CBAVD patients, the IVS9-Tn polymorphic locus was analyzed by two-round (nested) PCR. The results of PCR amplification and the allele-specific ligase reaction were evaluated by electrophoresis on an 8% polyacrylamide gel. The methods were previously described in detail [14,15]. Some variants were detected using additional molecular analysis of the *CFTR* gene using MLPA, DNA sequencing by the Sanger method, or massive parallel sequencing (MPS).

A classification of the genotypes into “severe” or “moderate” is based on the severity of the negative effect of different pathogenic *CFTR* variants on the function of the encoded protein in accordance with the generally accepted classification [3].

Statistical analysis was performed using the STATISTICA software, version 13.0 (StatSoft Inc., Tulsa, OK, USA). A significant difference in the frequencies of the *CFTR* gene alleles and genotypes was evaluated using the *χ*_2_ criterion. Results were considered reliable if the level of the index did not exceed 0.05 (*p* < 0.05).

## 5. Conclusions

Russian adult male patients with cystic fibrosis and CBAVD syndrome share a similar spectrum of pathogenic *CFTR* gene variants; however, some of the variants and genotypes dramatically differ between these patients’ groups. So, the “mild” CF-causing 3849+10kbC>T variant characterized for PS-CF is common in adult men with cystic fibrosis, including some azoospermic CF patients having CBAVD, but this variant is extremely rare in isolated congenital bilateral aplasia of the vas deferens. 394delTT is one of the common CF-causing variants in Russian CF patients, but not in CBAVD patients. In contrast, some other *CFTR* gene variants (L138ins, 2143delT, and W1282X) were detected 3–4 times more common in CBAVD, than in CF patients. Total frequency of CF-causing variants and 5T allele, as well as the structure of *CFTR* genotypes in Russian men with CBAVD syndrome, is very similar to the patients from European populations.

## Figures and Tables

**Figure 1 ijms-24-16287-f001:**
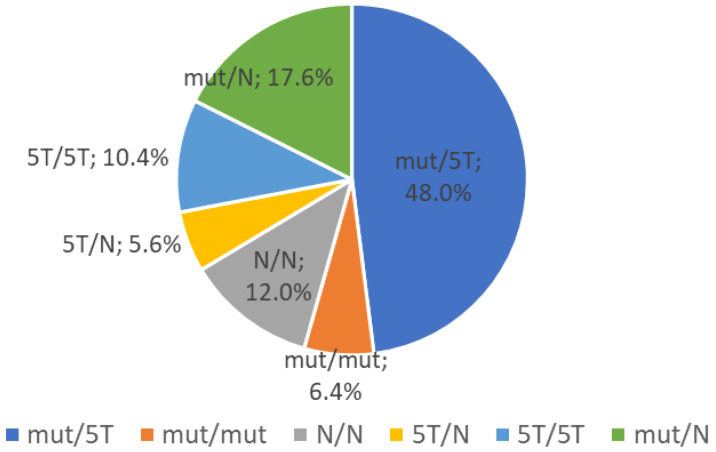
*CFTR* genotypes in Russian patients with CBAVD syndrome (*n* = 125).

**Table 1 ijms-24-16287-t001:** CF-causing *CFTR* gene variants in Russian men with CF and CBAVD syndrome.

Pathogenic *CFTR* Gene Variant	CF, *n* = 546		CBAVD, *n* = 125	
	Alleles, *n* (AF)	Percentage of Detected Variants, %	Alleles, *n* (AF)	Percentage of Detected Variants, %
c.1521_1523delCTT (p.Phe508del, F508del)	541 (0.49542) ^1^	52.7	50 (0.200) ^1^	51.0
c.54-5940_273+10250del21kb (p.(Ser18Argfs*16), CFTRdele2,3)	72 (0.65934) ^2^	7.0	7 (0.028) ^2^	7.1
c.3718-2477C>T (3849+10kbC-T)	55 (0.05037) ^3^	5.4	1 (0.004) ^3^	1.0
c.274G>A (p.Glu92Lys, E92K)	42 (0.03846)	4.1	4 (0.016)	4.1
c.2052_2053insA (p.Gln685ThrfsX4, 2184insA)	29 (0.02656)	2.8	4 (0.016)	4.1
c.2012delT (p.Leu671X, 2143delT)	23 (0.02106)	2.2	5 (0.020)	5.1
c.262_263delTT (p.Leu88IlefsX22, 394delTT)	21 (0.01923) ^4^	2.0	- ^4^	0.0
c.413_415dupTAC (p.Leu138dup; L138ins)	20 (0.01832)	1.9	8 (0.032)	8.2
c.3909C>G (p.Asn1303Lys, N1303K)	20 (0.01832)	1.9	3 (0.012)	3.1
c.1624G>T (p.Gly542X, G542X)	17 (0.01557)	1.7	3 (0.012)	3.1
c.3846G>A (p.Trp1282X, W1282X)	16 (0.01465)	1.6	6 (0.024)	6.1
c.1545_1546delTA (p.Tyr515X, 1677delTA)	12 (0.01099)	1.2	2 (0.008)	2.0
c.1000C>T (p.Arg334Trp, R334W)	9 (0.00824)	0.9	1 (0.004)	1.0
c.1397C>G (p.Ser466X, Ser466X)	8 (0.00733)	0.8	-	0.0
c.3844T>C (p.Trp1282Arg, W1282R)	8 (0.00733)	0.8	-	0.0
c.3140-16T>A (3272-16T>A)	8 (0.00733)	0.8	-	0.0
c.2657+5G>A (2789+5A>G)	6 (0.00549)	0.6	-	0.0
c.3691delT (p.Ser1231ProfsX4, 3821delT)	6 (0.00549)	0.6	-	0.0
c.1243_1247del (p.Asn415X, 1367del5)	6 (0.00549)	0.6	-	0.0
c.3196C>T (p.Arg1066Cys, R1066C)	4 (0.00366)	0.4	-	0.0
c.3587C>G (p.Ser1196X, S1196X)	4 (0.00366)	0.4	-	0.0
c.3816_3817delGT (p.Ser1273LeufsX28, 3944delGT)	3 (0.00274)	0.3	-	0.0
c.489+1G>T (621+1G>T)	3 (0.00274)	0.3	-	0.0
c.1766+1G>A (1898+1G>A)	3 (0.00274)	0.3	-	0.0
c.1766+1G>A (1898+1G>C)	3 (0.00274)	0.3	-	0.0
c.1040G>C (p.Arg347Pro, R347P)	3 (0.00274)	0.3	-	0.0
c.3484C>T (p.Arg1162X, R1162X)	3 (0.00274)	0.3	-	0.0
c.3476C>T (p.Ser1159Phe, S1159F)	3 (0.00274)	0.3	-	0.0
c.442del (p.Ile148LeufsX5, 574delA)	2 (0.00183)	0.2	-	0.0
c.472dupA (p.Ser158LysfsX5, 604insA)	2 (0.00183)	0.2	1 (0.004)	1.0
c.254G>A (p.Gly85Glu, G85E)	2 (0.00183)	0.2	1 (0.004)	1.0
c.3883delA; (p.Ile1295Phefs, 4015delA)	1 (0.00092)	0.1	1 (0.004)	1.0
c.3890_3891insT (p.Gly1298Trpfs, 4022insT)	1 (0.00092)	0.1	-	0.0
c.3140-26A>G (3272-26A>G)	1 (0.00092)	0.1	1 (0.004)	1.0
Others ^5^	69	6.7	-	0.0

^1^—the allelic frequency (AF) is significantly higher in CF patients (*χ*_2_ = 70.853; *p* < 0.001). ^2^—AF is significantly higher in CF patients (*χ*_2_ = 5.284; *p* = 0.022). ^3^—AF is significantly higher in CF patients (*χ*_2_ = 10.937; *p* < 0.001). ^4^—AF is significantly higher in CF patients (*χ*_2_ = 4.884; *p* = 0.028). ^5^—Other CF-causing variants: R553X, S945L, S1159P, 3849G>A,4382delA, Y569H, 175delC, 3659delC, 3849G>A, 574delA, and A141D (*n* = 22); 185+1G>T, 461A>G, 712-1G>T, 1248+1G>A, 1525-1G>A, 1716+1G->A, 2114delT, 2183AA>G, 2721del11, 2790-2A>G, 3667ins4, 4015delA, 4022insT, S945L, 624delT, 663delT, A96E, c.174_177delTAGA, c.1761del, c.3325delA, c.3893delG, c.4078delG, CFTRdele4-11(4-10*), CFTRdele19-21, D572N, G314R, G480S, L1335P, p.Asp993Ala, p.Glu402X, p.Glu1433Gly, p.Leu581X, p.Lys1468Asn, p.Pro988Arg, p.Tyr84X, Q98R, Q493R, R117C, R668C, R785X, R1158X, R1531I, T604I, W401X, W1310X, Y1032C, and IVS9-TG12T5 (*n* = 47).

**Table 2 ijms-24-16287-t002:** *CFTR* genotypes in Russian men with cystic fibrosis (*n* = 546).

*CFTR* Genotypes	Severity	Number, *n*	Percentage of Detected *CFTR* Genotypes, %
F508del/F508del	Severe	136	24.9
F508del/unknown	n.a.	45	8.2
F508del/3849+10kbC->T	Moderate	38	7.0
F508del/CFTRdele2,3	Severe	33	6.0
F508del/E92K	Moderate	23	4.2
F508del/2143delT	Severe	12	2.2
F508del/2184insA	Severe	12	2.2
F508del/L138ins	Moderate	9	1.6
F508del/G542X	Severe	8	1.5
F508del/W1282X	Severe	7	1.3
F508del/2789+5G>A	Moderate	5	0.9
F508del/N1303K	Severe	5	0.9
F508del/394delTT	Severe	4	0.7
F508del/3272-16T>A	Moderate	4	0.7
F508del/3821delT	Severe	4	0.7
CFTRdele2,3/2184insA	Severe	4	0.7
CFTRdele2,3/G542X	Severe	4	0.7
CFTRdele2,3/N1303K	Severe	4	0.7
E92K/E92K	Moderate	4	0.7
394delTT/394delTT	Severe	4	0.7
394delTT/unknown	n.a.	4	0.7
3849+10kbC>T/unknown	Moderate	3	0.5
CFTRdele2,3/L138ins	Moderate	3	0.5
E92K/N1303K	Moderate	3	0.5
F508del/1367del5	Severe	3	0.5
F508del/R334W	Moderate	3	0.5
F508del/R1066C	Severe	3	0.5
F508del/S1196X	Severe	3	0.5
F508del/1677delTA	Severe	2	0.4
F508del/3849G>A	Moderate	2	0.4
F508del/4382delA	Moderate	2	0.4
F508del/604insA	Severe	2	0.4
F508del/621+1G>T	Moderate	2	0.4
F508del/R1162X	Severe	2	0.4
F508del/W1282R	Moderate	2	0.4
F508del/Y569H	Moderate	2	0.4
F508del/1898+1G>A	Severe	2	0.4
CFTRdele2,3/CFTRdele2,3	Severe	2	0.4
CFTRdele2,3/1367del5	Severe	2	0.4
CFTRdele2,3/E92K	Moderate	2	0.4
CFTRdele2,3/R334W	Moderate	2	0.4
CFTRdele2,3/unknown	n.a.	2	0.4
3849+10kbC>T/S466X(TGA)/R1070Q	Moderate	2	0.4
2184insA/G542X	Severe	2	0.4
2184insA/L138ins	Moderate	2	0.4
2143delT/N1303K	Severe	2	0.4
2143delT/unknown	n.a.	2	0.4
1677delTA/1677delTA	Severe	2	0.4
394delTT/3272-16T>A	Moderate	2	0.4
W1282X/S466X(TGA)/R1070Q	Unknown	2	0.4
Others	-	112	20.5

n.a.—not assessed.

**Table 3 ijms-24-16287-t003:** *CFTR* genotypes in 125 Russian patients with CBAVD syndrome.

*CFTR* Genotypes		Number of Patients, N (Percentage of All Patients, %)
CF-Causing Variants	IVS-9Tn	
F508del/L138ins	9T/7T	4 (3.2%)
F508del/N1303K	9T/7T	1 (0.8%)
L138ins/N1303K	7T/9T	1 (0.8%)
L138ins/L138ins	7T/7T	1 (0.8%)
2184insA/E92K	7T/7T	1 (0.8%)
F508del/N	9T/5T	37 (29.6%)
	7T/9T	8 (6.4%)
L138ins/N	7T/5T	1 (0.8%)
CFTRdele2,3(21Kb)/N	7T/7T	2 (1.6%)
	7T/5T	5 (4.0%)
W1282X/N	7T/5T	5 (4.0%)
	7T7T	1 (0.8%)
2143delT/N	9T/5T	1 (0.8%)
	7T/5T	1 (0.8%)
	9T/9T	1 (0.8%)
	7T/9T	2 (1.6%)
2184insA/N	7T/5T	2 (1.6%)
	7T/7T	1 (0.8%)
E92K/N	7T/5T	1 (0.8%)
	7T/7T	1 (0.8%)
	7T/9T	1 (0.8%)
G542X/N	9T/5T	2 (1.6%)
	7T/5T	1 (0.8%)
N1303K/N	9T/5T	1 (0.8%)
1677delTA/N	7T/7T	1 (0.8%)
	7T/5T	1 (0.8%)
604insA/N	7T/5T	1 (0.8%)
3849+10KbC-T/N	7T/7T	1 (0.8%)
R334W/N	7T/7T	1 (0.8%)
4015delA/N	7T/7T	1 (0.8%)
3272-26A>G/N	7T/5T	1 (0.8%)
G85E/N	7T/5T	1 (0.8%)
N/N	9T/5T	2 (1.6%)
	5T/7T	5 (4.0%)
	5T/5T	13 (10.4%)
	7T/7T	11 (8.8%)
	7T/9T	4 (3.2%)

## Data Availability

Data are contained within the article.

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
