# Peer review of "Pathogenic Variants and Genotypes of the *CFTR* Gene in Russian Men with Cystic Fibrosis and CBAVD Syndrome"

_ijms, 2023, doi:10.3390/ijms242216287_

Round 1
Reviewer 1 Report
Comments and Suggestions for Authors
I would like to thank Authors for their study.
This paper describes a study aimed to analyze CFTR gene variants and genotypes in Russian men with Cystic Fibrosis and with CBAVD syndrome (azoospermia).
The results show differences in the frequency of some CFTR variants and genotypes in Russian CF and CBAVD patients; furthermore, among the CBAVD, eight subjects (6.4%) were found to have a genotype compatible with the diagnosis of Cystic Fibrosis.
This study may be interesting not only for CF specialists, moreover, andrologists, geneticists and molecular biologists could improve their knowledge on the importance of the CFTR gene in the pathogenesis of CBAVD syndrome.
However, I believe that some points of the paper need to be clarified
a) line 122 repeats what is reported in line 113
b) in table 2 there is a classification of the different genotypes, "severe" or "moderate". How was it determined? I assume it is based on the possible presence in patients of pancreatic sufficiency (moderate) or insufficiency (severe). Please add in the methods section on what severity is defined.
c) among CBAVD, with the study of the genotype, eight subjects (6.4% of the cohort) with two CF-causing variants were identified. In my opinion, this point needs greater depth in the discussion section, and if possible, it would be interesting to have some clinical information on these subjects (sweat test value, other symptoms if present, age) and on genetic and clinical counselling.
Author Response
Author's response to Reviewer 1.
We thank the Reviewer 1 for your careful consideration of the article, constructive criticism, comments and useful suggestions. The manuscript has been finalized according to the comments. All these deficiencies have been corrected.
Point 1: line 122 repeats what is reported in line 113
Response 1: The repetition in line 122 has been deleted. This paragraph (lines 119-123) has been corrected.
Point 2: in table 2 there is a classification of the different genotypes, "severe" or "moderate". How was it determined? I assume it is based on the possible presence in patients of pancreatic sufficiency (moderate) or insufficiency (severe). Please add in the methods section on what severity is defined.
Response 2: Actually, the classification of CFTR genotypes as "severe" or "moderate" was based on the severity of the negative effect of different pathogenic variants on the function of the encoded protein according to the generally accepted classification. Patients with pancreas sufficient CF (PS-CF) usually have moderate/mild genotypes, and patients with pancreas insufficient CF (PI-CF) usually have severe genotypes. This information has been added to the Methods section (lines 302-304).
Point 3: among CBAVD, with the study of the genotype, eight subjects (6.4% of the cohort) with two CF-causing variants were identified. In my opinion, this point needs greater depth in the discussion section, and if possible, it would be interesting to have some clinical information on these subjects (sweat test value, other symptoms if present, age) and on genetic and clinical counselling.
Response 3: These patients were examined at our center due to primary marital infertility associated with obstructive azoospermia. Some patients had signs of bronchopulmonary involvement in their history (recurrent bronchitis, sinusitis), but since CF was not suspected in them previously, a comprehensive evaluation was not performed and we do not have detailed clinical information on these patients. We recommended them to undergo a comprehensive examination, but now we do not have the results.

Reviewer 2 Report
Comments and Suggestions for Authors
The authors conducted a clinical study to investigate the frequency of pathogenic CFTR variants regarding the occurrence of CBAVD and CUAVD in Russian males with infertility. They analyzed genotypes and alleles from patients with CF and non-CF patients with infertility. They concluded the most frequent variant in CF and non-CF CBAVD patients is the F508del. Also, they found that the frequency of further CFTR variants results in CAVD diverging from other ethnic groups, and ethnicity-specific mutations are enriched in the Russian population. Furthermore, these studies highlighted, in agreement with other published data, that patients with non-CF CBAVD carry variants that would cause milder symptoms and are in the spectrum of the CF-RD category. The study carries essential information regarding the diagnosis of CBAVD and points out the importance of genetic testing prior to in vitro fertilization and planned parenthood.
Critics:
· Although the manuscript is well written, the references are used incorrectly in multiple places. Please crosscheck the references and what is stated in the manuscript.
1) Lines 37-38: “Male factor’ is revealed in about half of all infertile couples, with genetic abnormalities 37 are responsible for at least 50% cases of severe infertility [1].
This reference (1) discusses only female factors, which are present in about 35% of cases. So, the male factor should add up to 65% of the cases. Male infertility is not discussed in the here-referenced manuscript. Please define better where 50% of genetic abnormalities come from.
2) Lines 61-64: “Overall, in 80% patients, CBAVD syndrome is resulted from CF-causing variants/5T allele of the CFTR gene; rare cases of this disorder are caused by pathogenic variants in the ADGRG2 or other candidate genes (SLC9A3, SCNN1B, and CA12) [9].”
The referenced (9) manuscript does not make a statement regarding CBAVD. Please define where the statement was coming from.
3) Lines 65-66: “The CFTR gene is located on chromosome 7 (7q31.2), has a size of about 189 Kb, and consists of 27 exons and 26 introns [9].”
Reference [9] does not provide any information about the length and chromosomal location of the CFTR gene. Please use reference [8] for this statement.
4) Lines 68-69: “To date, more than 2,000 pathogenic variants in the CFTR gene have been identified [8].”
Please use reference [9] to this statement instead of reference [8].
5) Line 73: “…with the combination of two ‘severe’ pathogenic variants in cis-position”
I believe the authors want to use the trans-position instead of cis. Please double-check it.
· Line 27: Please define what N means in the genotype description. Like CFTRmut/N. I assume it means this allele is wild-type (no mutation detected) but should be defined.
Author Response
Author's response to Reviewer 2.
We would like to thank Reviewer 2 for your comments. The manuscript has been finalized according to the comments. All these deficiencies have been corrected. Thank you for taking the time to review our work.
Point 1: Lines 37-38: “Male factor’ is revealed in about half of all infertile couples, with genetic abnormalities 37 are responsible for at least 50% cases of severe infertility [1].
This reference (1) discusses only female factors, which are present in about 35% of cases. So, the male factor should add up to 65% of the cases. Male infertility is not discussed in the here-referenced manuscript. Please define better where 50% of genetic abnormalities come from.
Response 1: The reference was replaced.
Point 2: Lines 61-64: “Overall, in 80% patients, CBAVD syndrome is resulted from CF-causing variants/5T allele of the CFTR gene; rare cases of this disorder are caused by pathogenic variants in the ADGRG2 or other candidate genes (SLC9A3, SCNN1B, and CA12) [9].” The referenced (9) manuscript does not make a statement regarding CBAVD. Please define where the statement was coming from.
Response 2: The reference was replaced.
Point 3: Lines 65-66: “The CFTR gene is located on chromosome 7 (7q31.2), has a size of about 189 Kb, and consists of 27 exons and 26 introns [9].”
Reference [9] does not provide any information about the length and chromosomal location of the CFTR gene. Please use reference [8] for this statement
Response 3: The reference was replaced.
Point 4: Lines 68-69: “To date, more than 2,000 pathogenic variants in the CFTR gene have been identified [8].” Please use reference [9] to this statement instead of reference [8].
Response 4: The reference was replaced.
Point 5: Line 73: “…with the combination of two ‘severe’ pathogenic variants in cis-position”. I believe the authors want to use the trans-position instead of cis. Please double-check it.
Response 5: Yes, you are absolutely right, of course “… two pathogenic variants in trans-position”.
Point 6: Line 27: Please define what N means in the genotype description. Like CFTRmut/N. I assume it means this allele is wild-type (no mutation detected) but should be defined.
Response 6: Unlike СF patients (who must have two mutated CFTR alleles a priory), CBAVD patients may have one or two 'normal' CFTR alleles. If they have no (detected) mutation/5T allele, it is usually regarded as ‘normal’, and designed by ‘N’ in the genotype.
